# Molecular Signatures in Ductal Carcinoma In Situ (DCIS): A Systematic Review and Meta-Analysis

**DOI:** 10.3390/jcm12052036

**Published:** 2023-03-03

**Authors:** Drissa Ouattara, Carole Mathelin, Tolga Özmen, Massimo Lodi

**Affiliations:** 1Surgery Department, Point G University Hospitals, Bamako P.O. Box 251, Mali; 2Strasbourg University Hospital, 1 Avenue Molière, 67200 Strasbourg, France; 3Surgical Oncology Department, ICANS Institute of Oncology Strasbourg Europe, 17 Avenue Albert Calmette, CEDEX, 67200 Strasbourg, France; 4IGBMC Institute of Genetics, Molecular and Cellular Biology, CNRS, UMR7104 INSERM U964, Strasbourg University, 1 Rue Laurent Fries, 67400 Illkirch-Graffenstaden, France; 5Division of Gastrointestinal and Oncologic Surgery, Department of Surgery, Massachusetts General Hospital, Harvard Medical School, 55 Fruit St, Boston, MA 02114, USA

**Keywords:** ductal carcinoma in situ, molecular assay, radiotherapy, local recurrence, de-escalation, precision medicine

## Abstract

Context: Adjuvant radiotherapy (RT) after breast-conserving surgery (BCS) for ductal carcinoma in situ (DCIS) is debated as benefits are inconstant. Molecular signatures for DCIS have been developed to stratify the risk of local recurrence (LR) and therefore guide the decision of RT. Objective: To evaluate, in women with DCIS treated by BCS, the impact of adjuvant RT on LR according to the molecular signature risk stratification. Methodology: We conducted a systematic review and meta-analysis of five articles including women with DCIS treated by BCS and with a molecular assay performed to stratify the risk, comparing the effect of BCS and RT versus BCS alone on LR including ipsilateral invasive (InvBE) and total breast events (TotBE). Results: The meta-analysis included 3478 women and evaluated two molecular signatures: Oncotype Dx DCIS (prognostic of LR), and DCISionRT (prognostic of LR and predictive of RT benefit). For DCISionRT, in the high-risk group, the pooled hazard ratio of BCS + RT versus BCS was 0.39 (95%CI 0.20–0.77) for InvBE and 0.34 (95%CI 0.22–0.52) for TotBE. In the low-risk group, the pooled hazard ratio of BCS + RT versus BCS was significant for TotBE at 0.62 (95%CI 0.39–0.99); however, it was not significant for InvBE (HR = 0.58 (95%CI 0.25–1.32)), Discussion: Molecular signatures are able to discriminate high- and low-risk women, high-risk ones having a significant benefit of RT in the reduction of invasive and in situ local recurrences, while in low-risk ones RT did not have a benefit for preventing invasive breast recurrence. The risk prediction of molecular signatures is independent of other risk stratification tools developed in DCIS, and have a tendency toward RT de-escalation. Further studies are needed to assess the impact on mortality.

## 1. Introduction

Ductal carcinoma in situ (DCIS) accounts for approximately 15.2% of all breast cancers in the United States [1]. Moreover, its incidence has increased in the last decades, in part as a consequence of the generalization of breast cancer screening, but also due to population aging and increased risk factors such as obesity or family history [2,3]. Of interest, DCIS is a component of *BRCA* spectrum cancers, and *BRCA2* carriers have higher incidences in DCIS compared to *BRCA1* carriers [4], although DCIS is equally as prevalent in *BRCA* mutation carriers as in high familial-risk noncarrier women [4,5]. According to the World Health Organization (WHO), DCIS are a heterogeneous group of lesions characterized by a proliferation of ductal neoplastic cells confined within the breast duct. They are distinguished from invasive carcinomas by the absence of visualization of infiltration of the breast stroma and the respect of the basal membrane [6]. True prevalence in the general population is unknown and varies with age. Some autopsy series found DCIS in 9–14.7% of women dying from other causes [7,8]. Currently, DCIS is mostly asymptomatic and diagnosed by radiological examination in 86% of cases [9], particularly in the form of microcalcifications alone (75–85%) [9,10] or as a linear or segmental nonmass enhancement at breast magnetic resonance imaging [11]. However, some cases present a palpable mass, a nipple discharge or a Paget’s disease of the nipple [3,12].

DCIS’s natural history is not yet completely understood. It has been shown that it is a non-obligate precursor lesion of invasive breast carcinoma [7,13]. Women with untreated DCIS have an increased risk of developing invasive breast cancer compared to women diagnosed with non-proliferative lesions (odds radio = 13.5) [14]. More recently, a 30 years follow-up study of untreated low-grade DCIS showed that invasive breast cancer developed in 36% of cases and this evolution can occur after several decades [15]. Similarly, some small historical series of untreated DCIS (initially misdiagnosed as a benign lesion) showed that the development of invasive breast cancer ranged from 20% [16] to 53% [17].

The main objective of DCIS treatment is to prevent progression to the invasive form, as DCIS itself has no theoretical metastasis and mortality risks. Currently, overdiagnosis and overtreatment in DCIS is a hot topic and commonly discussed in the scientific era [18]. Treatment options depend on different factors such as the clinical and radiological presentation, national guidelines, physician and patient preferences. They include mastectomy with or without immediate breast reconstruction or breast conserving surgery (BCS) with or without radiation therapy. These local treatments may be accompanied by adjuvant endocrine therapy [19]. While surgical treatment with clear margins is well established, the role of endocrine therapies is still controversial and did not demonstrate a benefit in overall survival and may lead to morbidity and adverse events [19,20,21]. Radiotherapy was once mandatory, and more recently its indications and modalities have evolved [19,20,21]. The standard of care consists of whole breast radiotherapy, typically within 8 weeks after BCS. Radiotherapy modalities are heterogeneous, as this treatment may be with or without a tumor bed boost (if risk factors of recurrence are present) [22], and also because of new protocols such as hypofractionated schedules [23,24], or accelerated partial-breast irradiation [25].

Different DCIS characteristics have been studied to stratify the risk, such as the grade, hormonal receptor and HER2 expression, and presence of comedonecrosis [26]. However, even if invasive recurrence rates are higher for high-risk compared to low-risk DCIS, those characteristics are insufficient to define a subgroup where it is oncologically safe to avoid adjuvant therapy. Recently, molecular assays have been developed for DCIS risk stratification and decision making in adjuvant radiotherapy [27], and their use is increasing in DCIS management [28]. Two molecular assays are currently available: Oncotype© DX DCIS test and DCISionRT (Prelude Dx, Laguna Hills, CA, USA) [26]. Oncotype DX DCIS evaluates 12 genes expression by RT-PCR on surgical specimens and provides information on the 10-year invasive or in situ local recurrence after BCS, irrespective of endocrine therapy and radiotherapy treatment [29]. This test was later refined by incorporating additional clinical and pathological features such as age and tumor size [30]. On the other hand, DCISionRT combines the expression of different molecular biomarkers and different clinical and pathological features and is able to determine prognosis (risk of invasive and in situ local recurrence) and to predict the benefit of radiotherapy for patients who have undergone BCS [31]. As new data emerged, scientific literature is heterogeneous on these molecular assays for DCIS [13]. Therefore, the aim of this study was to evaluate through a systematic review of the literature and a meta analysis the impact of adjuvant radiotherapy on local recurrence (invasive and non-invasive disease) in women with DCIS treated with BCS according to their molecular assay risk stratification.

## 2. Materials and Methods

We adhered to the Preferred Reporting Items for Systematic Reviews and Meta-Analyses (PRISMA) guidelines [32].

### 2.1. Eligibility Criteria

Prospective and/or retrospective clinical trials and/or cohort studies were included if they met the following eligibility criteria:Population: women with DCIS treated by BCS and with a molecular assay performed to stratify the risk.Intervention: no radiotherapy (BCS alone).Comparator: radiotherapy (BCS + RT).Outcome: local recurrence (LR), which is defined as ipsilateral breast event (InvBE) or ipsilateral total breast event (TotBE, including both invasive and in situ local recurrences.)

We included only articles where a follow-up was available. If more than one study involved the same population, only the most recent study or the one with the highest number of cases was included in the analysis.

### 2.2. Bibliographic Selection

The initial query was performed on 23 June 2022 on PubMed and Scopus and included the following keywords: “ductal carcinoma in situ” or “dcis”; “multigene” or “gene”; “oncotype dx dcis” or “dcisionrt” or “dcis score”. The initial query gave 684 results. These articles were analyzed by two independent reviewers (DO and ML). Based on the title and abstract, 638 articles were excluded because they were not directly related to the subject under study, because of an unassessed follow-up, or because they were meta-analyses, correspondence, literature reviews, basic research articles, animal, or in vitro studies. We kept 10 articles that were selected for full-text review. Among those, we excluded 5 articles because they did not investigate LR according to risk group and treatment. We kept 5 articles for qualitative synthesis and for the meta-analysis (quantitative synthesis); however, one of them reported LR risk only for BCS treatment alone without comparator. Discrepancies between the 2 reviewers were resolved by consensus. The bibliographic selection, with exclusion reasons, is reported in the flow chart (Figure 1).

### 2.3. Data Collection

For each article, both reviewers (DO and ML) independently extracted the following information: first author name, year of publication, type of multigene assay, the hazard ratio (HR) and its 95% confidence interval (95%CI) of BCS + RT versus BCS alone according to each risk group and the number of patients in each risk group. In addition, country, years of inclusion, principal results and adjusting variables were retrieved.

### 2.4. Statistical Analysis

Analysis was performed separately between the two molecular signatures, as DCISionRT is a predictive and prognostic test while Oncotype Dx DCIS is a prognostic test. Moreover, the levels of evidence required for the two are different and the other clinical assays that can be used as comparators are also different. For each article, we compared the difference between oncological outcomes (TotBE and InvBE) between two groups: BCS alone versus BCS + RT. Analysis was stratified by risk group (high versus low) and by the molecular signature. The meta-analysis was performed using R version 4.1.3 (10 March 2022) [33] and with the *metafor* package [34]. Given the heterogeneity of the populations in our different studies, the random effect model was used in the meta-analysis. For each article, we calculated the natural logarithm of the hazard ratio (HR), and the standard error. The articles were weighted on the standard error. We calculated pooled HR with an estimated 95% confidence interval (95%CI); and heterogeneity was quantified with a maximum-likelihood estimator for $\tau^{2}$ and we calculated the Higgins $I^{2}$ statistic. For the test of heterogeneity, the Cochran Q *p*-value was obtained with a Wald-type test.

## 3. Results

We selected five articles [30,35,36,37,38] that included a total of 3478 women treated for DCIS (described in Table 1): 1485 for DCISionRT; and 1993 for Oncotype Dx DCIS. For DCISionRT, three articles evaluated the 10-year recurrence risks according to the interventions (standard treatment/omission of radiotherapy) [35,36,37]. For Oncotype Dx DCIS, one article evaluated the 10-year recurrence risks according to the interventions (standard treatment/omission of radiotherapy) [38] after stratification of the patients into low- and high-risk groups, and one article evaluated the 10-year recurrence risk for patients with BCS without radiotherapy according to the molecular signature risk group and two clinical and pathological features [30]. Absolute 10-year risk of local recurrence is reported in Table 2 and Table 3 for DCISionRT and Oncotype Dx DCIS, respectively. Absolute risks were stratified by treatment (BCS alone versus BCS + RT) and type of breast event (both in situ and invasive versus invasive).

### 3.1. DCISionRT

#### 3.1.1. Qualitative Synthesis

Bremer et al. [36] conducted a cohort study including women from Uppsala University Hospital and Västmanland County Hospital, Sweden (UUH) and the University of Massachusetts, Worcester (UMass), and evaluated DCISionRT between 1986 and 2008 (retrospective assay). They included 526 women, 216 with BCS alone and 310 with BCS + RT. They found a significant reduction in the 10-year risk of recurrence for patients in the high-risk group, whether the recurrence was invasive or in situ (hazard ratios 0.3 (0.1–0.6) and 0.3 (0.1–0.5); *p*-values 0.003 and <0.001, respectively). Conversely, in the low-risk group, the benefit of RT was not significant (*p*-value = 0.305). Moreover, in a subgroup of women with favorable clinicopathological characteristics, 42% of them were reclassified as high risk according to the molecular signature and had a significant benefit of RT compared to BCS alone.

Wärnberg et al. [37] investigated DCISionRT in the SweDCIS randomized trial between 1987 and 2000, comparing BCS alone versus BCS + RT for DCIS (retrospective assay). Among the trial participants, the authors included 504 women with complete data and negative margins in DCISionRT evaluation. The results of this study showed that a significant benefit of radiotherapy was present in the high-risk group (defined as a decision score > 3) for both total and invasive local recurrences (respectively, HR = 0.32 (95%CI 0.17–0.58, *p*-value < 0.001) and HR = 0.24 (95%CI = 0.08–0.74, *p*-value = 0.013)). Conversely, in the low-risk group (decision score ≤ 3), the benefit of RT was not significant for the total (HR = 0.53 (95%CI = 0.28–1.02, *p*-value = 0.059) and invasive local recurrences (HR = 0.84 (95%CI 0.30–2.31, *p*-value = 0.73)).

Weinmann et al. [35] conducted a study including 455 women from the Kaiser Permanente Northwest Tumor Registry from 1990 to 2007 and evaluated DCISionRT (retrospective assay). The included women had two different treatments: BCS alone (n=79) versus BCS + RT (n=377). First, they found that after adjustment on treatment group, the decision score was associated with total and invasive local recurrence risk. In the BCS + RT group, the high-risk group compared to the low-risk group, the HR of total and invasive local recurrences were 1.72 (95%CI = 0.86–3.41) and 1.73 (95%CI = 0.74–4.05), respectively. In the BCS alone group, these risks were greater (nearly 2-fold): 3.04 (95%CI = 0.95–9.73) and 3.80 (95%CI = 0.76–0.18.92), although not statistically significant. These observations suggest that the RT benefit is greater for the high-risk group, both on in situ and invasive local recurrences.

#### 3.1.2. Meta-Analysis: Absolute Risk of Local Recurrence

For the DCISionRT high-risk group, the total breast event absolute risk ranged from 23 to 30% in the case of BCS alone and 8.3 to 11% in the case of BCS + RT, with an absolute risk reduction of 12–20% with the addition of RT. The invasive breast event risk ranged from 12.4 to 21% in the case of BCS alone and 3.1 to 9% in the case of BCS + RT, with an absolute risk reduction of 6 to 15% with the addition of RT. In the DCISionRT low-risk group, the total breast event absolute risk ranged from 8 to 12.9% in the case of BCS alone and 5 to 7.2% in the case of BCS + RT, with an absolute risk reduction of 1 to 5.7% with the addition of RT. The invasive breast event risk ranged from 5 to 7.7% in the case of BCS alone and 3 to 6.5 in the case of BCS + RT, with an absolute risk reduction of 1 to 2% with the addition of RT.

#### 3.1.3. Meta-Analysis: Hazard Ratio of Local Recurrence for DCISionRT

As different cohorts were available for DCISionRT, we calculated the pooled hazard ratio of local recurrence (both total and invasive) according to the risk group and to the treatment type, as reported in Figure 2. In the high-risk group, the pooled hazard ratio of BCS + RT versus BCS was 0.39 (95%CI 0.20–0.77) for invasive breast events and 0.34 (95%CI 0.22–0.52) for total breast events, showing a significant benefit of RT in the reduction of invasive and in situ local recurrences. Conversely, in the low-risk group, the pooled hazard ratio of BCS + RT versus BCS was significant for the total breast events at 0.62 (95%CI 0.39–0.99); however, it was not significant for invasive breast events (HR = 0.58 (95%CI 0.25–1.32)), showing that RT did not have a benefit of preventing invasive breast recurrence.

### 3.2. Oncotype DX DCIS

#### 3.2.1. Qualitative Synthesis

Rakovitch et al. [38] evaluated Oncotype DX DCIS on an Ontario DCIS cohort between 1994 and 2003 with a mean follow-up of 9.4 years (retrospective assay). The patients were divided into two groups according to their treatment: 571 with BCS alone and 689 with BCS + RT. The authors found that women with a low-risk score treated by BCS alone had a small benefit from RT (not statistically significant) compared to those with a high score having a greater benefit from RT (statistically significant). Moreover, they found that a subgroup of women with a high score and favorable clinicopathological characteristics had a higher risk of local recurrence and greater benefit of RT than the control group (investigated further in a subsequent article [30]).

Rakovitch et al. conducted this supplementary investigation in another article [30]. They included patients from two cohorts: the Ontario DCIS cohort (including only women with BCS alone) and the ECOG-ACRIN E5194 cohort (which evaluated BCS surgery without radiation + Tamoxifen versus BCS alone without radiation [39]). They included 773 women with DCIS treated by BCS alone to investigate clinicopathological characteristics able to refine the DCIS score risk stratification (retrospective assay). They found that age (with a cut-off of 50-years-old) and tumor size (with a cut-off of 2.5 cm), combined with a decision score, predicted better situations of low (defined as ≤8%) or high (defined as >15%) 10-year local recurrence risk after BCS compared to a decision score or clinicopathological characteristics alone.

It must be noted that this signature has three different risk groups: high, low and intermediate. In the intermediate risk group, the decision of RT omission may be more complicated and thus leads to a limitation of the use of this molecular signature in DCIS.

#### 3.2.2. Meta-Analysis: Absolute Risk of Local Recurrence

The results for Oncotype DX DCIS were similar, although only one article reported data of the 10-year absolute risk of total breast events. For the high-risk group, the absolute risk was 32.7% in the case of BCS alone and 20% for BCS + RT, hence a reduction of 12.7% of the risk. For the low-risk group, the absolute risk was, respectively, 16% and 9.4%, and the risk reduction was 6.6%. Moreover, another study showed that this risk can be adjusted according to two clinicopathological features: age (inversely proportional) and tumor size (proportional). No data for the absolute risk of invasive breast events was found for Oncotype Dx DCIS.

## 4. Discussion

In this systematic review and meta-analysis, we wanted to evaluate the impact of adjuvant radiotherapy on local recurrence (invasive and non-invasive) in women with DCIS treated by BCS and a DCIS molecular assay risk stratification. Interestingly, the results showed that molecular signatures were able to stratify the risk of local recurrence and therefore to predict the benefit of radiotherapy. Indeed, in women with high-risk DCIS, radiotherapy significantly reduced the risk of invasive and non-invasive local recurrence. Conversely, in the low-risk DCIS group, radiotherapy did not significantly reduce the risk of invasive local recurrence (but reduced the risk of non-invasive local recurrence). Therefore, the question of de-escalation by omitting adjuvant radiotherapy seems legitimate in DCIS patients with low risk molecular signatures.

### 4.1. DCIS Prognosis and Radiotherapy after BCS

DCIS has an excellent prognosis, and to date is considered stage 0 breast cancer according to the 8th edition of the Union for International Cancer Control (UICC) and American Joint Committee on Cancer (AJCC) classification. Indeed, in a study including more than 100,000 women diagnosed with DCIS from 1988 to 2011 in the Surveillance, Epidemiology, and End Results (SEER), it has been shown that breast cancer-specific mortality was 3.3% (95%CI, 3.0–3.6%) at 20 years from diagnosis [40]. This risk was higher if the age at diagnosis was <35 years and if an ipsilateral invasive breast cancer occurred. Nonetheless, there were 517 patients (0.48%) who died of breast cancer following a DCIS diagnosis without experiencing a *local* invasive recurrence. Interestingly, radiotherapy did not impact 10-years breast cancer mortality in women diagnosed with DCIS treated with BCS (0.8% versus 0.9%; HR, 0.86 (95%CI, 0.67–1.10); *p*-value = 0.22). Similarly, other publications found that radiotherapy did not show a benefit in mortality reduction, although it reduced the risk of local recurrence. In a meta-analysis published in 2007 of four randomized clinical trials including 3665 women with DCIS treated by BCS, the authors found a significant reduction of invasive and in situ ipsilateral breast cancer with adjuvant radiotherapy; however, there was no risk reduction of distant metastases and breast-cancer mortality [41]. Moreover, this study showed an 1.5-fold increased risk of contralateral breast cancer after radiotherapy [41]. The authors suggested that the method of delivery of RT in some studies may explain these findings. Then, in 2010, another meta-analysis was published, including 3729 women with DCIS treated by BCS. This study found a significant reduction in the absolute 10-year risk of any ipsilateral breast event, regardless of the age at diagnosis, tamoxifen, surgical margins, focality and pathological parameters such as focality, grade, size, comedonecrosis and focality [21]. However, this study did not show a benefit of radiotherapy in breast-cancer specific and overall mortality. Finally, in 2018, another meta-analysis was published including long-term data from four randomized clinical trials including 3680 women with DCIS treated by BCS [42]. Again, the authors found that radiotherapy significantly reduced the ipsilateral breast and regional recurrence risks, without impacting the distant recurrence risk and overall mortality. Conversely, it has been shown that prior radiotherapy increases breast-cancer mortality in case of invasive local recurrence HR = 1.70 (1.18–2.45) *p* = 0.005 [43]. In addition, radiotherapy treatments may cause a significant morbidity with acute (such as skin changes, breast induration, pain and cosmetic alterations) and late toxicities (such as heart and lung) that must be taken into account in the benefit-risk balance. For these reasons, radiotherapy benefits are still debated in DCIS management for women treated by BCS. Data from the literature suggest that some women may indeed benefit from radiotherapy after BCS for DCIS. However, this treatment seems insufficient without further discrimination, as showed by previous results. The difficulty therefore lies in discriminating between a subgroup of these women who may be at high risk of local recurrence and in whom radiotherapy presents more benefits than risks. For these reasons, different risk-stratification tools have been developed within this context, aiming to find the balance between over- and undertreatment.

### 4.2. Risk Stratification in DCIS

In an interesting systematic review published by Schmitz et al. in 2022 [44], the authors reported different published models predicting breast events after DCIS. The first published models were based on clinicopathological factors. The Van Nuys Prognostic Index (VNPI) was the first one published in 1996 by Silverstein et al. [45], and included tumor size, margin width, and pathologic classification of the DCIS. This score aimed to identify a low-risk of recurrence population, in which an omission of radiotherapy could be chosen. In 2010, Rudloff et al. [46] published the Memorial Sloan Kettering Cancer Center (MSKCC) nomogram, a tool integrating 10 clinicopathological factors (see Table 4), which estimates the 5 and 10-year ipsilateral breast event (in situ or invasive) in women with DCIS treated with BCS. Finally, in 2016 Sagara et al. [47] published a new score, which aimed to predict the survival benefit of radiotherapy after DCIS treated by BCS. This score was based on clinico-pathological characteristics (see Table 4).

The molecular signatures for DCIS are more recent, and were published in 2013 for Oncotype DX DCIS [29] and in 2018 for DCISionRT [36]. Their approach is different as they are not based on solely clinicopathological features available from pathology reports (and therefore their utilization needs further biological assays), but from supplementary analyses on tumor specimens. Oncotype DX DCIS is a multigene assay that evaluates 12 genes’ expression by RT-PCR on surgical specimens and provides information on the 10-year invasive or in situ local recurrence after BCS, irrespectively of endocrine therapy and radiotherapy treatment [29]. A later publication implemented this tool with a refined DCIS Score (RDS) that additionally incorporates three clinico-pathological factors (see Table 4) to improve the model performances [49]. Of interest, the 21-gene Oncotype Dx recurrence score, indicated for invasive breast cancer, has also been evaluated in DCIS recently [48]. This study was conducted on 1362 women ≤ 75 years-old with DCIS treated with BCS ± RT. The authors found that the recurrence score combined with age could identify women for whom radiotherapy reduces the risk of 20-year breast cancer-specific mortality. In addition, in a recent study comparing the VNPI, the MSKCC nomogram and the Oncotype DX DCIS score, Lei et al. found that differences exists between the estimated ipsilateral breast event risks, suggesting that these models are not interchangeable [50].

Taken together, these data suggest that molecular signatures may have a significant benefit in clinical practice, as they are independent from other clinico-pathological-based models, and can efficiently predict the risk of local recurrence and the local benefit of radiotherapy. Still, the existent data were insufficient on the benefit on mortality, which is an essential parameter for estimating the true benefit of radiotherapy.

### 4.3. Molecular Signatures Significantly Impact Radiotherapy Decision and Economic Costs

Different studies evaluated the impact of these molecular signatures in the decision of adjuvant RT and cost effectiveness of genomic testing in DCIS. For Oncotype DX DCIS, Alvarado et al. showed that this molecular signature changed adjuvant RT decisions in 31% of the cases, and globally led toward a RT de-escalation (73% of adjuvant RT pre-assay versus 59.1% post-assay, *p*-value = 0.008). Moreover, the authors reported that physicians rated the molecular signature as the most impactful factor in planning treatment [51]. Similarly, Shah et al. evaluated DCISionRT and found that adjuvant RT recommendations decreased by 20% [52]. The elevated risk determined by the molecular signature had the strongest association with an RT recommendation (odds ratio 43.4) [52]. Finally, a cost effectiveness study was conducted on DCISionRT and showed that “molecular signature for all patients and RT for elevated risk only” was cost-effective compared with the “no molecular signature, RT for all” strategy when the cost of DCISionRT was less than $4588 [53]. Another study showed that DCIS management using DCISionRT testing was a cost-effective strategy in terms of quality-adjusted life-years saved compared to a traditional strategy without a molecular signature [54]. These data suggest that including molecular signatures in DCIS management after BCS could significantly change clinical practices in the long term without increasing public health costs. Moreover, given that these signatures have a tendency to RT de-escalation, they may lead to less acute and late radiation toxicities and therefore improve the quality of life for women with DCIS. Nonetheless, there are other de-escalation possibilities currently under investigation, such as “active” surveillance without surgery.

### 4.4. Future Perspectives in Treatment De-Escalation for DCIS

Overtreatment is central to the current debate on the management of DCIS. The development of molecular signatures is in line of a more global tendency in DCIS management: the treatment de-escalation. Indeed, without locoregional treatment (i.e., surgery and radiotherapy) the 10-year risk of invasive cancer is 10.5% and breast-cancer specific mortality is 3.9% [55]. Moreover, the benefit of treatment depends strongly on the age at diagnosis. Currently, several clinical trials are conducted worldwide to assess de-escalation possibilities in DCIS such as the LORIS [56] (United Kingdom), LORD [57] (Europe), COMET [58] (United States), ROMANCE [59] (France) and the LORETTA [60] (Japan) trials. These clinical trials aim to investigate not only RT de-escalation, but also surgery omission (i.e., active surveillance or endocrine therapy alone). In these trials, the definition of low-risk DCIS is based on clinical and pathological features. Consequently, the approach of molecular signatures is significantly different, as surgery cannot be avoided with this strategy. To date, no results of these trials are yet published and therefore molecular signatures remain the sole option toward RT de-escalation in DCIS treated with BCS.

### 4.5. Limitations of This Study

This study was conducted according to international methodology guidelines and to our knowledge this is the first meta-analysis on this subject. Moreover, the results are consistent between the two available molecular signatures and therefore support their validity in clinical practice. However, these results must be put into perspective because of certain limitations. First, no data were available on breast-cancer specific mortality. Then, individual data on recurrence-free survival were not available and therefore a survival meta-analysis could not be performed. The data presented are absolute risk, and this does not allow for an assessment of risk at each time point but only at 10 years, so information on early and later prognosis is not available.

## 5. Conclusions

The goal of treatment for DCIS is to prevent the development of invasive cancer while avoiding overtreatment of patients. This study shows that, for women with DCIS treated with BCS, molecular signatures can provide an accurate local recurrence risk assessment, and in particular they can predict the benefit of RT, and this independently of other prediction models, which are based on clinical and pathological characteristics. Indeed, in the low risk groups, RT has a low benefit, while in the high risk group, RT can reduce the risk of invasive recurrence. In addition, the utilization of these signatures in clinical practice has a tendency towards RT de-escalation. The scientific literature suggests that molecular signatures are a promising tool for finding the balance between over- and undertreatment. Still, further understanding of DCIS and the basis of invasion is needed. Different basic research projects (such as the PRECISION initiative [61]) are ongoing, and may in the future improve risk quantification and identify new targets to prevent invasion.

## Figures and Tables

**Figure 1 jcm-12-02036-f001:**
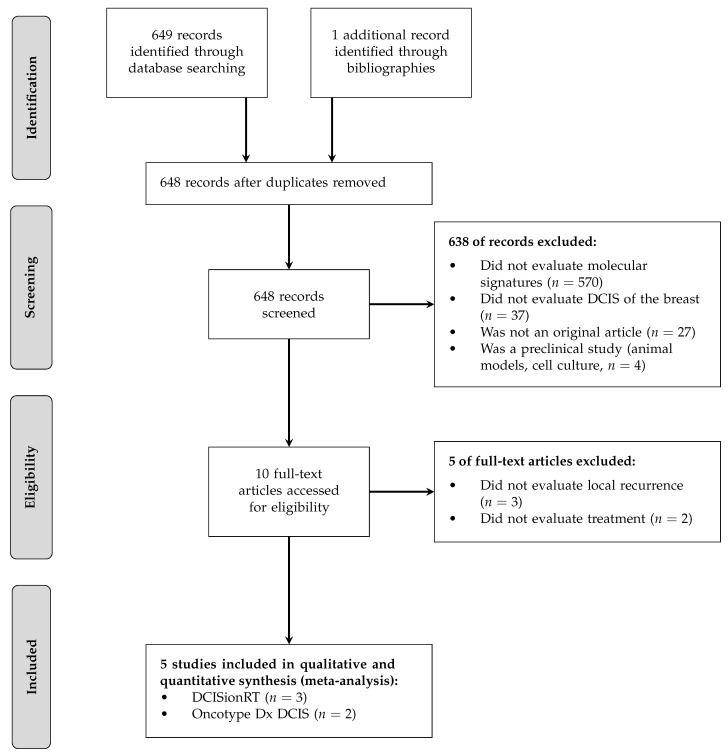
Flow chart diagram (PRISMA). Legend: DCIS = ductal carcinoma in situ; BCS = breast-conserving surgery.

**Figure 2 jcm-12-02036-f002:**
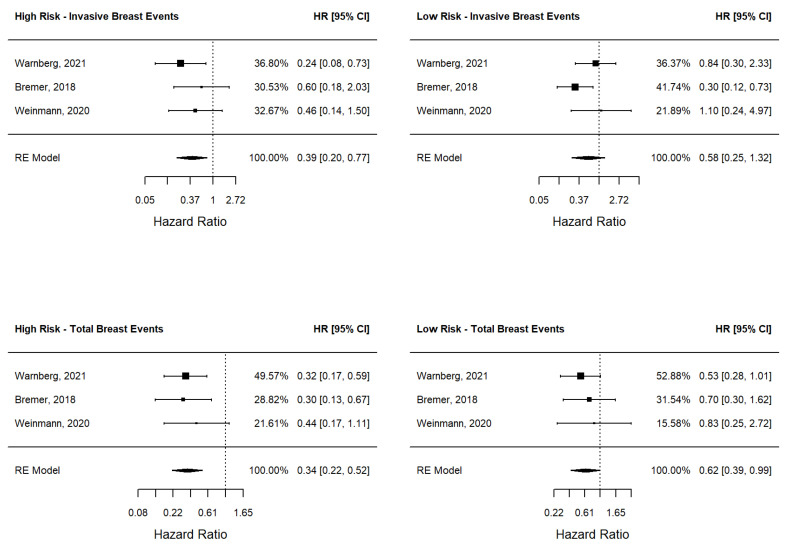
Meta-analysis of DCISionRT studies: Forest plots. Legend: HR = hazard ratio; 95%CI = 95% confidence interval; RE model = random effect model; Percentages represent the individual weight of each study, this measure is calculated according to the sample size and its heterogeneity [35,36,37].

**Table 1 jcm-12-02036-t001:** Summary of included studies with GRADE rating of the certainty of evidence.

Author, Year	Study Design	Risk of Bias	Inconsis-Tency	Indirect-Ness	Impreci-Sion	Other Considerations	Summary of Findings	Certainty
Weinmann, 2020 [35]	cohort, retrospective and prospective (follow-up: median 10.4 years)	not serious	not serious	serious ${}^{a}$	not serious	strong association; all plausible residual confounding would reduce the demonstrated effect	Years: 1990–2007; Country: United States; Total patients: 455 (BCS alone *n* = 78; versus BCS + RT *n* = 377); Molecular signature: DCISionRT; Test result: decision score (DS); Outcome: 10-year local recurrence; Main findings: High risk group (DS > 3): HR 2.03 (1.12–3.70) for TotBE and 2.14 (1.00–4.59) for InvBE (reference: low-risk group)	Moderate
Bremer, 2018 [36]	cohort, retrospective (follow-up: median 9 years)	not serious	not serious	serious ${}^{a}$	not serious	strong association; all plausible residual confounding would reduce the demonstrated effect	Years: 1986–2004/1999–2008 (2 centers); Country: United States and Sweden; Total patients: 526 (BCS alone *n* = 216; versus BCS + RT *n* = 310); Molecular signature: DCISionRT Test result: decision score (DS); Outcome: 10-year local recurrence; Main findings: The High Risk group received significant RT benefit, with HR of 0.3 (0.1–0.6) for InvBE and 0.3 (0.1–0.5) for TotBE. In the Low Risk group these benefits were not significant (0.6 [0.2–2.3] and 0.7 [0.3–1.6], respectively).	Moderate
Warnberg, 2021 [37]	cohort, retrospective (follow-up: median 17.1 years)	not serious	not serious	serious ${}^{a}$	not serious	strong association; all plausible residual confounding would reduce the demonstrated effect	Years: 1987–2000; Country: Sweden; Total patients: 502 (BCS alone *n* = 247; versus BCS + RT *n* = 257); Molecular signature: DCISionRT; Test result: decision score (DS) Outcome: 10-year local recurrence; Main findings: significative risk reduction for RT in the High Risk group (both total and invasive local recurrence, respectively, HR 0.32 (0.17–0.58) and HR 0.24 (0.08–0.74). In the Low Risk group, there were no significant risk differences observed with RT.	Moderate
Rakovitch, 2017 [38]	cohort, retrospective (follow-up: median 9.4 years)	not serious	not serious	serious ${}^{a}$	serious ${}^{b}$	strong association; all plausible residual confounding would reduce the demonstrated effect	Years: 1994–2003; Country: Canada; Total patients: 1260 (BCS alone *n* = 571; versus BCS + RT *n* = 689); Molecular signature: Oncotype DX DCIS; Test result: Decision Score; Outcome: 10-year local recurrence; Main findings: The DS risk group was statistically significantly associated with LR risk (HR high/intermediate 1.75 (1.28–2.41). Women with a low-risk DS treated by BCS alone had an LR risk of 10.6% at 10 years and a small benefit from RT, while those with a high DS had a higher risk of LR (25.4%) after BCS alone and greater benefit from RT.	Low
Rakovitch, 2018 [30]	cohort, retrospective (follow-up: median 11.5 years)	not serious	not serious	serious ${}^{a}$	serious ${}^{b,c}$	strong association; all plausible residual confounding would reduce the demonstrated effect; dose response gradient	Years: *N.A.*; Country: United States; Total patients: 773 (BCS alone *n* = 773); Molecular signature: Oncotype DX DCIS; Test result: decision score (DS) + tumor size + age; Outcome: 10-year local recurrence; Main findings: Women with Low DS, age ≤ 50 and tumor size ≤ 1 cm had a low risk of local recurrence (7.2% [5.3–10.0]) while women with High DS, age < 50 and tumor size ≥ 2.5 cm had a high risk of local recurrence (8.6 [44.1–66.5]). Utilization of DS combined with tumor size and age at diagnosis predicted more women with very low (≤8%) or higher (>15%) 10-year LR risk after BCS alone compared to the utilization of DS alone or clinicopathological factors alone.	Moderate

^*a*^ mortality is not assessed; ^*b*^ presence of an intermediate risk group, ^*c*^ no control group.

**Table 2 jcm-12-02036-t002:** DCISonRT 10-year absolute risk of local recurrence stratified by treatment, risk group and recurrence type.

Molecular Assay	Author, Year	Women (n)	Risk Group	Treatment	10-Year Total BE Absolute Risk (%) [95%CI]	Difference	10-Year Invasive BE Absolute Risk (%) [95%CI]	Difference
DCISionRT	Weinmann, 2020 [35]	455	High	BCS + RT	10 [6–15]	20	6 [3–10]	15
				BCS alone	30 [17–51]		21 [9–44]	
			Low	BCS + RT	5 [2–10]	5	3 [1–9]	2
				BCS alone	10 [3–29]		5 [1–30]	
	Bremer, 2018 [36]	526	High	BCS + RT	11 [4–17]	12	9 [3–15]	6
				BCS alone	23 [11–33]		15 [5–24]	
			Low	BCS + RT	7 [1–13]	1	3 [0–7]	1
				BCS alone	8 [0–14]		4 [0–9]	
	Warnberg, 2021 [37]	504	High	BCS + RT	8.3 [4.5–15.3]	15.5	3.1 [1.2–8.1]	9.3
				BCS alone	23.8 [14.8–36.8]		12.4 [7.2–20.8]	
			Low	BCS + RT	7.2 [3.5–14.6]	5.7	6.5 [3.2–13.2]	1.2
				BCS alone	12.9 [6.9–23.5]		7.7 [3.9–14.9]	

Legend: 95%CI = 95% confidence interval; BE = breast event (local recurrence); BCS = breast conservative surgery;
RT = radiotherapy.

**Table 3 jcm-12-02036-t003:** Oncotype Dx DCIS 10-year absolute risk of local recurrence stratified by treatment, risk group and recurrence type.

Molecular Assay	Author, Year	Women (n)	Risk Group	Treatment	10-Year Total BE Absolute Risk (%) [95%CI]	Difference	10-Year Invasive BE Absolute Risk (%) [95%CI]	Difference
Oncotype DX DCIS	Rakovitch, 2017 [38]	1260	High	BCS + RT	20 [15.9–24.9]	12.7	NA	NA
				BCS alone	32.7 [25.9–40.6]		NA	
			Low	BCS + RT	9.4 [7–12.5]	6.6	NA	NA
				BCS alone	16 [12.2–20.9]		NA	
Oncotype DX DCIS + CPF	Rakovitch, 2018 [30]	733	High	BCS alone	14.6 [12.9–23.1] T < 1, A ≥ 50	NA	NA	NA
					48.6 [44.1–66.5] T ≥ 2.5, A < 50		NA	
			Low	BCS alone	7.2 [5.3–10.0] T < 1, A ≥ 50	NA	NA	NA
					30.2 [20.6–36.1] T ≥ 2.5, A < 50		NA	

Legend: 95%CI = 95% confidence interval; BE = breast event (local recurrence); BCS = breast conservative surgery;
RT = radiotherapy; CPF = clinical and pathological features (patient age and tumor size); NA = not available;
T = tumor size (cm); A = age (years).

**Table 4 jcm-12-02036-t004:** Risk stratification tools in DCIS.

Name	Type	Clinical/Pathological Features	Biological Assay
VNPI ^1^ [45]	Score (1–3)	Tumor size, margin width and pathologic classification	None
MSKCC nomogram ^2^ [46]	Nomogram (0–100)	Age at diagnosis, family history, initial presentation (clinical/radiologic), radiation, adjuvant endocrine therapy, nuclear grade, necrosis, margins, number of excisions, year of surgery	None
Patient Prognostic Score [47]	Score (0–6)	Age at diagnosis, tumor size, nuclear grade	None
DCISionRT [35,36,37]	Molecular assay	None	7 proteins (COX-2, FOXA1, HER2, Ki-67, p16/INK4A, PgR, and SIAH2)
Oncotype DX DCIS [29,38]	Molecular assay	None	12 genes: 7 cancer-related (Ki-67, AURKA/STK15, BIRC5/survivin, CCNB1, MYBL2, PGR, and GSTM1) and 5 reference (ACTB, GAPDH, RPLPO, GUS, and TFRC)
Refined Oncotype DX DCIS [30]	Molecular assay and score	Age, tumor size	12 genes: 7 cancer-related (Ki-67, AURKA/STK15, BIRC5/survivin, CCNB1, MYBL2, PGR, and GSTM1) and 5 reference (ACTB, GAPDH, RPLPO, GUS, and TFRC)
Oncotype DX [48]	Molecular assay	None	21 genes: 16 cancer-related (Ki-67, AURKA/STK15, BIRC5/survivin, CCNB1, MYBL2, MMP11, CTSL2, GRB2, HER2, ER, PGR, BCL2, SCUBE2, GSTM1, CD68 and BAG1) and 5 reference (ACTB, GAPDH, RPLPO, GUS, and TFRC)

^1^ Van Nuys Prognostic Index; ^2^ Memorial Sloan Kettering Cancer Center.

## Data Availability

Data sharing is not applicable to this article as no new data were created or analyzed in this study.

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
