# Peer review of "Molecular Signatures in Ductal Carcinoma In Situ (DCIS): A Systematic Review and Meta-Analysis"

_jcm, 2023, doi:10.3390/jcm12052036_

Round 1

Reviewer 1 Report

In this manuscript  authors did a meta analysis trying to define the impact  molecular signatures have in radiotherapy decision after BCS for DCIS.

After  a search in the literature and a double blind analysis they based their study in five articles including 3478 women with DCIS treated with BCS with or without RT.

The focus of the manuscript is an important issue in clinical practice and this has been very well described in introduction. The manuscript is clearly written and the results are significant.

Author Response

Dear Reviewer, 

We thank you for your review and your positive feedback. 

Reviewer 2 Report

The authors performed a meta-analysis of women with Ductal Carcinoma in situ (DCIS) treated by breast-conservative surgery to determine the benefit of additional radiotherapy.  They evaluated two genetic tests (Oncotype DXDCIS, DCISionRT) to identify women who would benefit from radiotherapy.

The study is carefully designed and executed.  The authors cite 61 publications of which 26 are from the last three years.  The conclusions drawn are well supported by the data.

Author Response

(The authors gave the same response as above.)

Reviewer 3 Report

The authors undertake a systematic review the molecular signatures of DCIS and provide their thoughts supported by data. Themanuscript is well written and data clearly presented. 

One major concerns I have in the manner in which data is presented is that their appears to be comparison between DCIS Score and DecisionRT. This may not be completely a correct way of doing the analysis as one is "prognostic" assay and while DecisionRT is a predictive assay. Having these presented slide by side in the same table gives the impression of direct comparison. 

One would recommend dealing with them separately as the levels of evidence required for the 2 are different and the other clinical assays that can be used as comparitors are different. I understand this will be in some ways tearing the paper apart but it will be scientifically more accurate.  

Author Response

Dear Reviewer, 

First of all, we would like to thank you for your high-quality review and comments. Ideed, as you pointed out, the presentation of the results appears to compare DCISionRT and Oncotype Dx DCIS, which was not our intention. Consequently, we revised the manuscript structure to separate the two molecular signatures. 

Manuscript changes include : 

  • a statement in the abstract
  • a separation of the two signatures in the flow chart (figure 1)
  • a new paragraph at the beginning of the section 2.4 Statistical analysis
  • Former Table 2 was separated in Table 2 and 3 according to the molecular signature
  • A new Result subsection structure with 3.1 DCISionRT (including qualitative synthesis, absolute risk meta-analysis and pooled HR meta-analysis) and 3.2 Oncotype Dx DCIS (with qualitative synthesis, absolute risk meta-analysis)

We think that this modification has improved the quality of the manuscript and the scientific accuracy as you suggested.